# Toxic Elements and Mineral Content of Different Tissues of Endemic Edible Snails (*Helix vladika* and *H. secernenda*) of Montenegro

**DOI:** 10.3390/foods9060731

**Published:** 2020-06-03

**Authors:** Vesna Vukašinović-Pešić, Bogumiła Pilarczyk, Tymoteusz Miller, Monika Rajkowska-Myśliwiec, Joanna Podlasińska, Agnieszka Tomza-Marciniak, Nada Blagojević, Nevzeta Trubljanin, Andrzej Zawal, Vladimir Pešić

**Affiliations:** 1Faculty of Metallurgy and Technology, University of Montenegro, 81000 Podgorica, Montenegro; nadab@ucg.ac.me; 2Department of Animal Reproduction Biotechnology and Environmental Hygiene, Faculty of Biotechnology and Animal Husbundry, West Pomeranian University of Technology, 71-415 Szczecin, Poland; Bogumila.Pilarczyk@zut.edu.pl (B.P.); Agnieszka.Tomza-Marciniak@zut.edu.pl (A.T.-M.); 3Institute of Marine and Environmental Sciences, University of Szczecin, 71-415 Szczecin, Poland; tymoteusz.miller@usz.edu.pl (T.M.); Andrzej.Zawal@usz.edu.pl (A.Z.); 4Department of Toxicology, Dairy Technology and Food Storage, Faculty of Food Science and Fisheries, West Pomeranian University of Technology, 71-415 Szczecin, Poland; Monika.Rajkowska@zut.edu.pl; 5Department of Environmental Management, Faculty of Environmental Management and Agriculture, West Pomeranian University of Technology, 71-415 Szczecin, Poland; Joanna.Podlasinska@zut.edu.pl; 6Department of Biology, University of Montenegro, 81000 Podgorica, Montenegro; trubljanin.nena@t-com.me (N.T.); vladopesic@gmail.com (V.P.)

**Keywords:** mineral content, toxic elements, accumulation, edible snails, snail tissue, Montenegro

## Abstract

The objective of the present study is to determine the differences between the mineral content of various organs of *Helix vladika* and *H. secernenda*, the two most abundant edible snail species in Montenegro. The bioaccumulation of 12 examined elements (zinc, manganese, copper, aluminum, cadmium, lead, nickel, iron, chromium, lithium, selenium and mercury) was determined in the hepatopancreas, albumen gland, digestive tract, reproductive system, mantle, foot and shell from three sampling sites (Biogradska Gora, Nikšić and Malesija). The examined populations of *H. vladika* and *H. secernenda* showed a significant difference in their lithium and selenium contents. The levels of the most examined metals (Zn, Mn, Cu, Al, Cd, Pb, Se and Hg) varied significantly among organs. The digestive tract and hepatopancreas tend to bioaccumulate selenium and cadmium. The general mean concentration of cadmium in the examined snail tissues exceeded the maximum allowable level at the Biogradska Gora and Malesija sites. Therefore, the use of the Montenegrin edible snails collected from the wilderness for human consumption seems to be limited by their higher bioaccumulation capacity for toxic elements such as cadmium.

## 1. Introduction

Snail meat has long been recognized as a high-quality food. Snail meat contains high amounts of protein, a relatively low lipid content [1] and a balanced intake of omega-3 and omega-6 fatty acids that have led to its recognition, due to its nutritional and therapeutic benefits [2,3]. On the other hand, land snails are considered as appropriate sentinel species [4,5] because trace metals tend to accumulate in their tissues, particularly in the digestive glands [6]. The edible snails of the genus *Helix* could be a powerful tool for environmental pollution assessment [3]. Depending on the soil mineral content and level of contamination, the snails can accumulate large quantities of heavy metals such as lead and cadmium, therefore making the snail meat a very high-risk food product [7]. 

In Montenegro, the harvesting of edible snails has boomed in the first decade of the 20th century. Based on the data from the International Heliciculture Institute from Cherasco (Italy), Montenegro has 22 snail farms with 7 hectares [8] established before 2010. All of these farms use the extensive (the Italian) method with outdoor breeding, but in practice, they are the most commonly worked with populations of snails collected in the wild that are bred for a while and then exported. At that time, many farmers were attracted by the idea of getting almost immediate benefits. However, after 2010, inadequate legislation as well as poor management in snail farming led to almost all farmers having given up on snail farming, what led to a quick failure of this snail farming practice and the disappearance of almost all farms in Montenegro. 

Several species of the genus *Helix* are present in Montenegro, among them the most widespread and abundant are *Helix vladika* Kobelt, 1898, and *H. secernenda* Rossmässler, 1847. The latter species is with wide distribution from the Croatian coastland to Northern Greece, while *H. vladika* is endemic to Montenegro, but also found in the neighboring parts of Serbia and Albania. Due to its larger size and morphological resemblance to *H. pomatia*, which is the most widely used edible snail in Europe, *H. vladika* is more intensely exploited in Montenegro. 

It is known that edible snails can accumulate higher concentrations of some essential elements such as selenium (Se), copper (Cu) and zinc (Zn), and therefore can be beneficial for human nutrition. Many studies have mentioned snail meat as a source of selenium for human health and nutrition [2,7]. Most of these studies (e.g., [2,3,7,9,10,11,12]) have been done on the Roman snail (*Helix pomatia*), which is the most widely used edible snail species in Europe, while other species are very poorly researched and little is known (or unknown as is the case with our species) about the mineral content of their meat. 

This study aims to determine the contents of zinc, manganese (Mn), copper, aluminum (Al), cadmium (Cd), lead (Pb), nickel (Ni), iron (Fe), chromium (Cr), lithium (Li), selenium and mercury (Hg) in different snail tissue of the two most abundant edible snails of the genus *Helix* in Montenegro. Statistical analyses were conducted to establish possible correlations in the element content distribution with different snail tissues between species and sampling localities. The obtained data are allowed to identify different snail tissues as a potential source of some essential elements for human nutrition. On the other hand, this study was aimed also to evaluate the risk assessment of the accumulation of toxic elements such as Pb, Cd and Hg which are considered dangerous and priority pollutants by the EU legislation (European Parliament and Council of European Union, Directive 2000/60/EC).

## 2. Materials and Methods 

### 2.1. Material

The study was performed on two species of edible snails of the genus *Helix*, i.e., *H. vladika* and *H. secernenda*, that inhabit the territory of Montenegro. The snails were collected from three sampling sites: (1) Nikšić, Dragovoljići village (42°46′19.48″ N, 19°2′30.65″ E), (2) Malesija, Skorać village (42°22′2.83″ N, 19°23′20.30″ E) and (3) Biogradska Gora (42°53′39.54″ N, 19°36′17.20″ E). The Biogradska Gora site where *H. vladika* was collected is located in a protected area, the Biogradska Gora National Park. Two other sites, Malesija (Skorać) and Nikšić (Dragovoljići), are inhabited by *H. secernenda* and are located outside the motorized traffic, and were not affected by any source of pollution. Samples were taken in May (at Biogradska Gora and Nikšić) and in September 2018 (Malesija). From each site, 30 snail specimens with almost a similar size were randomly taken. In addition, three subsamples of soils were randomly taken (at a depth of 10 cm) from each site. Snails were put in plastic bags, transferred in the laboratory and thawed at room temperature on a clean tissue paper. The soft tissues were then separated from the shell, and dissected and pooled into six different parts, namely the mantle, hepatopancreas, digestive tract, albumen glad, reproductive system and foot. The ethical guidelines have been strictly followed in dealing with experiments on animals using the minimum number to obtain a significant result of the experimental work. For the allometric studies, five specimens from each site were selected and measured for the following parameters: shell height, shell weight, tissue wet weight, tissue dry weight and water content (Table 1). 

### 2.2. Chemical Analysis

After dissection, the tissue samples were dried at 60 °C for 72 h in sterile Petri dishes to constant dry weights. The soil sediment samples were dried at 60 °C for at least 72 h to a constant dry weight. 

The concentrations of Pb and Cd were analyzed by means of graphite furnace atomic absorption spectrometry with Zeeman background correction (GF-AAS, 4110 ZL Perkin Elmer) and other elements (Al, Fe, Mn, Zn, Cu, Cr, Ni, Li) using inductively coupled plasma atomic emission spectrometry equipped with the Meinhard TR 50-C1 nebulizer (ICP-AES, Yobin YvonJY-24). 

The samples were weighed in an amount of 0.2 g ± 0,001 g into the Teflon^®^ vessels of a microwave oven (CEM MDS 2000). As the oxidizing agent, 5 mL of 65%, analytical grade HNO_3_ (Merck KGaA, Darmstadt, Germany) was added per sample. The mineralization process proceeded in 5 steps under the conditions shown in Table 2. The digested samples were filtered into volumetric flasks and topped up to 25 mL using bidistilled water.

The calibration curve technique was used for the quantification of toxic elements. Single-element calibration standards (Merck, Germany) were used as the stock standards for preparing the working standards. Working standards were prepared by serial volume/volume dilution in glass volumetric flasks with 0.2% HNO_3_ (Pb, Cd) or 15% HNO_3_ (others elements) in Milli-Q water (Merck, Darmstadt, Germany). The linearity was considered satisfactory and acceptable if the correlation coefficient, r, exceeded 0.999, and in the case of Pb and Cd, we found a correlation coefficient of r^2^ ≥ 0.9997. For quality control, blanks, calibration standards of each analyte and standard reference materials were tested during each batch.

The limits of detection (LOD) were determined and calculated based on the data from the calibration curve [13]. The calculated LODs were as follows (μg/L): Al 50.0, Fe 4.0, Mn 1.4, Zn 1.8, Cu 5.4, Cr 8.92, Ni 5.76, Li 2.0, Pb 1.04 and Cd 0.098. 

In order to evaluate the accuracy of the methods, the following certified materials were used: MODAS-2, Bottom Sediment (M–2 BotSed) or MODAS–3, Herring Tissue (M–3 HerTis) and 2 blank samples. The results obtained by means of the GF-AAS and ICP-AES methods were compared with the certified values, and recoveries were calculated (Table 3).

Total Hg (THg) concentrations were determined using atomic absorption spectroscopy (AAS). The assays were run in an AMA 254 mercury analyzer (Altach Ltd., Czech Republic). The AMA 254 analyzer allowed the determination of Hg in samples without having to perform prior mineralization in wet conditions. The detection limit for this device is 0.01 ng THg. The limit of detection was calculated based on three times the standard deviation of the blank. A blank (i.e., an empty sample nickel boat) was analyzed periodically to verify that mercury was not being carried over between samples.

The analytical procedure was checked by determining THg concentrations in samples of one reference material CRM015-050 (metals on sediment analyte concentrations; *n* = 3). The recovery rate was 99.4%.

The Se concentrations in samples were determined using spectrofluorometric methods (Shimadzu RF–5001 PC, Shimadzu Corporation, Tokyo, Japan). The samples were digested in HNO_3_ (Chempur, Poland) at 230 °C for 180 min and in HClO_4_ (Chempur, Poland) at 310 °C for 20 min. Then, selenate (Se^6+^) was reduced to selenite (Se^4+^) using 3 mL of 9 % HCl (Chempur, Poland). EDTA and hydroxylamine hydrochlorine were used as masking agents. Then, selenium was derivatized with 2,3-diaminonaphthalene (Sigma Aldrich, USA), under the conditions of controlled pH (pH 1–2) with the formation of a selenodiazole complex. Fluorescence of the organic (cyclohexane, Chempur, Poland) layer was measured using an emission wavelength of 518 nm and an excitation wavelength of 378 nm. Blank samples were determined concurrently with the samples proper. The LOD was 0.003 μg/g. The accuracy of the analyses was verified using certified reference material NIST SRM 1946 (Lake Superior Fish Tissue). Recovery ranged from 93% to 98% of the reference value. The precision (RSD %) of the analysis was 2.9%. All samples were analyzed in triplicate and their average value was assessed. 

### 2.3. Statistical Analysis

All statistical analyses were performed using the Statistica 13.0 PL software. Data were expressed as mean ± standard deviation. The bioaccumulation factor (BAF) was calculated on a dry weight basis, as BAF = concentration of the target element in the snail tissue/concentration of the same element in the ground.

The normal distribution in each group was tested with the Kolmogorov–Smirnov test, applicable for small samples. Analyses of variance (ANOVA) were computed to test if the observation depended on one or several factors acting simultaneously.

When the assumption of homogenous variances was fulfilled, the post-hoc Tukey’s *t*-test was applied to compare the groups for statistically significant differences. In the case when variances were heterogenous, a Kruskal–Wallis test was used. Differences were considered as significant at the level of *p* < 0.05. The Mann–Whitney test was used to investigate whether the mineral contents varied significantly between the studied snail species and between sampling seasons. 

Pearson correlation analysis was applied to examine the relationship between the concentrations of the studied metals in the selected organs and ground. 

In calculating the canonical discriminant functions, the forward method was performed in order to examine the discrimination power properties of variables in discrimination of the investigated tissue. Discriminant analysis (DA) was run on the so-called raw data, introducing appropriate organ grouping. When assessing the received orthogonal canonical functions, the comparison of our own values and the ƛ-Wilks test were used. On the basis of the first two discriminatory functions, the coordinate system of the canonical values was plotted to illustrate the grouping of these values.

## 3. Results and Discussion

### 3.1. Distribution of Element Concentrations in Studied Edible Snail Species of Montenegro

The investigated species are the most widespread edible snails in Montenegro [14], and therefore important for human consumption. Regarding the growing interest in edible snails as a potential group of alternative foods, the determination of the factors that affect the content of metals is gaining in importance. The allometric characteristics of the investigated populations of the two analyzed edible snail species, *Helix vladika* and *H. secernenda*, are given in Table 1. 

The mean concentrations of the analyzed minerals and toxic elements in the tissues of the studied snail populations from three sampling sites in Montenegro, together with bioaccumulation factor (BAF), are reported in Table 4. The Mann–Whitney test revealed a significant difference between *Helix vladika* (Biogradska Gora) and *H. secernenda* (Nikšić and Malesija) only in terms of the content of Li (*p* = 0.006) and Se (*p* = 0.002).

In general, the concentration of the most tested metals was lower at the Biogradska Gora site inhabited by *Helix vladika* and highest at the Malesia site inhabited by *H. secernenda* (see Table 4). However, for the general mean metal concentration in all snails, no significant difference between the studied localities were found (*p* = 0.7563). The reason for this may be that the examined localities in our study were not affected by any kind of pollution. In the study of Ćirić et al. [3], the metal content in the examined tissues of *H. pomatia* from four different environment locations in Pančevo city in Serbia were significantly higher at the polluted sites.

It is well known that metal content variability is not only affected by the possible level of pollution, but also by the geochemical characteristics of the habitat that snails inhabit as well as the season influence [2,3]. In our study, the canonical analysis performed on the metals variability at the investigated sampling sites separated the Malesia locality (see Figure 1a). On the other hand, the resulting scatter plot of the first and second canonical scores shows a part overlap of the Biogradska Gora and Nikšić localities, suggesting that these two sites share similarities in metal content variability. The first discriminant function explains 40% of the total variance and it is dominated by the Fe content (R = 0.99). The second discriminant function explains 25% of the total variance and it is dominated by the Zn content (R = 0.98). The variable with the greatest ability for discrimination along the first discriminant was Fe, while Zn plays an important role in the discrimination along the second discriminant function.

Samples from Biogradska Gora (*H. vladika*) and Nikšić (*H. secernenda*) were taken in May while snails from Malesija (*H. secernenda*) were collected in September, which may have an impact on the metal content in the snail issue. A number of studies have shown that the maximum metal content in snail tissue is reached in autumn and winter and the lowest levels in summer [3,15,16]. In the study of Ćirić et al. [3], the highest concentrations of Cu, Cd, Zn, Fe and Mn in snail tissues were found in October and the lowest in June. In our study, however, the Mann–Whitney test did not reveal a significant difference in seasonal sampling between the studied populations, neither between *H. secernenda* populations from Malesija and Nikšić.

### 3.2. Distribution of Element Concentrations in Different Snail Tissues

The metal accumulation in the different tissues of the snails of the genus *Helix* depends on the investigated snail tissue [3,9]. In our study, we found significant differences in the concentrations of the studied elements between the ground and examined tissues (Table 4). With the exception of Ni, Fe, Cr and Li, all other studied elements showed significant differences in the metal content between the analyzed snail tissues. (Table 4). The comparison of each metal concentration in tissue at subsequent sampling sites revealed significant differences in the Zn (*p* = 0.032), Cu (*p* = 0.017) and Cd (*p* = 0.039) contents. The BAF values for Zn from 1 to 2.3, indicate a bioaccumulation of this metal in the hepatopancreas.

The highest concentrations of Zn, Ni, Mn, Pb, Cd and Hg were found in the hepatopancreas. The concentration of Zn in the hepatopancreas was highest in snails from Nikšić (387.0 ± 16.9) and lowest in snails from Biogradska Gora (151.6 ± 6.9 mg kg^−1^ d.w.). The highest concentration of Zn in the hepatopancreas is in agreement with similar studies (e.g., [2,11]) on the metal content in snail tissues of *Helix pomatia.*

The concentration of Ni in the hepatopancreas ranged from 0.82 ± 0.12 in snails from Nikšić to 2.12 ± 0.36 mg kg^−1^ d.w. in snails from Malesija. On the other hand, the concentration of Mn in the latter tissue was highest in snails from Nikšić (405.12 ± 11.0) and lowest in snails from Malesija (32.4 ± 0.3 mg kg^−1^ d.w.). The BAF values for Ni and Mn below 1 (Table 4) indicate that they do not bioaccumulate in snail tissues.

The Pb concentration in the hepatopancreas ranged between 0.99 ± 0.29 (Malesija) to 5.46 ± 0.18 mg kg^−1^ d.w. (Biogradska Gora). Further, the Cd concentration in the latter tissue was highest in snails from Malesija (16.38 ± 1.46 mg kg^−1^ d.w.) and lowest in snails from Nikšić (1.43 ± 0.03 mg kg^−1^ d.w.). The BAF values for Pb were below 1, indicating that it did not bioaccumulate in the snail tissues. On the other hand, the high BAF values for Cd, ranging from 1.2 (Nikšić) up to 25.2 (Malesija), indicate the bioaccumulation of this element in the hepatopancreas.

The Hg concentration varied from 0.02 ± 0.0008 in snails from Nikšić to 0.054 ± 0.0021 mg kg^−1^ d.w. in snails from Biogradska Gora. The Fe, Cr, Al, Li and Se contents were highest in the digestive tissue. The concentration of Fe in the latter tissue was highest in snails from Malesija (805.0 ± 67.6) and lowest in snails from Biogradska Gora (111.9 ± 11.1 mg kg^−1^ d.w.). The Cr concentration was lowest in snails from Biogradska Gora (1.15 ± 0.58) and highest in the digestive tissue of snails from Malesija (4.84 ± 0.43 mg kg^−1^ d.w). Similarly, concentrations of Al and Li were highest in Malesija (1411.1 ± 172.0 and 1.26 ± 0.07 for Al and Li, respectively) and lowest in Biogradska Gora (73.6 ± 8.0, 0.03 ± 0.05 mg kg^−1^ d.w. for Al and Li, respectively). As can be seen from Table 4, the BAF values for Hg, Fe, Cr, Al and Li below 1 indicate that they did not bioaccumulate in the snail tissues.

The selenium concentration in the digestive tissues varies from 0.106 ± 0.02 in snails from Biogradska Gora to 0.587 ± 0.015 mg kg^−1^ d.w. in snails from Malesija. The high BAF values for Se from 2.2 (Biogradska Gora) to 9.6 (Malesija) tend for a bioaccumulation of this element in the digestive tract. The general mean value of selenium in the examined snail tissues of *H. vladika* (Biogradska Gora) was 0.04 mg kg^−1^ wet weight (w.w.) and from 0.05 to 0.08 mg kg^−1^ w.w. for *H. secernenda* (Nikšić and Malesija). These values were lower in comparison with those reported for the raw meat of Roman snails from Poland (0.09 mg kg^−1^) [2] as well as from those reported for the meat of *H. pomatia* snails from Moldavia, Ukraine and Russia, which varied in the range between 0.130 and 0.423 mg kg^−1^ [7]. Nevertheless, in our study, the Se concentration varies among snail tissues as well as between the investigated sites, reaching the highest value of 0.587 ± 0.015 mg kg^−1^ d.w. in the digestive tissue of snails from Malesia. Based on the available literature, the highest Se content in snail meat was found in *H. pomatia* from Moldavia and was 0.423 mg kg^−1^ [7].

The general mean values of selenium in our study were comparable with the content of the latter element in the mealworm (*Tenebrio molitor*) larvae, which ranged from 0.057 to 0.085 mg kg^−1^ w.w. [17]. The values of the Se content in the snails of our study (0.04–0.08 mg kg^−1^ w.w.) were similar with those reported for pork (0.078 mg kg^−1^ w.w.) and beef (0.064 mg kg^−1^ w.w.) [18], as well as for the liver (0.06 mg kg^−1^ w.w.), kidney (0.41 µg/g w.w.), heart and lungs (0.05 mg kg^−1^ w.w.) of roe deer [19].

On the other hand, our values for selenium were lower in comparison with those reported for wild boars and fishes. The concentration of selenium was 0.19 mg kg^−1^ w.w. in the liver and 1.20 mg kg^−1^ w.w in the kidney of wild boars from the northwest part of Poland [20]. In fish, the concentration of selenium varies normally between 0.2 and 0.9 mg kg^−1^ [21,22,23], reaching the highest values in species that are at the top of the food chain such as swordfish and tuna [24,25]. The level of selenium in the bluefin tuna (*Thunnus thynnus*) caught in the Mediterranean area varies from 0.6 ± 0.3 mg kg^−1^ (wild population) to 1.1 ± 0.9 mg kg^−1^ w.w. in farmed specimens [25]. Therefore, the studied populations of edible snails in Montenegro can be considered as a poor source of selenium for human consumption.

The most Cu was accumulated in the reproductive organ tissue. The highest Cu amount (98.4 ± 4.3) was found in snails from Nikšić and the lowest in snails from Biogradska Gora (53.9 ± 6.5 mg kg^−1^ d.w.). The BAF values, from 1.5 to 2.5, indicate a bioaccumulation of this metal. Dallinger and Wiesser [9] showed that the distribution of Cu in a snail tissue of *H. pomatia* is relatively even, without a clear preference for any organ.

The lowest concentration of the studied metals, with the exception of Ni, Al, Li and Pb, were found in the shells. The concentration of metals in the shells varied: for Zn, from 3.7 ± 0.1 (Malesija and Nikšić) to 4.2 ± 0.3 (Biogradska Gora); Fe, from 2.8 ± 0.8 (Biogradska Gora) to 26.8 ± 4.7 (Nikšić); Mn, from 2.3 ± 1.0 (Malesija) to 6.9 ± 1.3a (Biogradska Gora); Cr, from 0.26 ± 0.15 (Biogradska Gora) to 0.50 ± 0.17 (Malesija); and for Cu, from 7.8 ± 1.8 (Biogradska Gora) to 12.9 ± 9.4 mg kg^−1^ d.w. (Nikšić).

The lowest concentration of Ni and Pb were found in the mantle tissue. The concentration of Ni and Pb were higher in the mantle tissue of snails from Malesija (0.42 ± 0.25 and 0.21 ± 0.04 for Ni and Pb, respectively) in comparison with snails from Nikšić (0.13 ± 0.03 and 0.01 ± 0.00 mg kg^−1^ d.w., for Ni and Pb, respectively). The lowest concentration of Al was found in the albumen gland tissue of snails from Nikšić (8.6 ± 1.2), and the lowest concentration of Li was found in the foot tissue of snails from Biogradska Gora (0.03 ± 0.05 mg kg^−1^ d.w.).

Our study confirmed previous studies that species of the genus *Helix* can be used as a bioindicator of a heavy metal accumulation, since they concentrate metals in their tissues [26]. The comparative analysis of the mean values shows that the accumulation of metals in the studied snail populations has the following order: Al > Fe > Mn > Zn > Cu > Cr > Li > Ni > Pb > Cd > Se > Hg. The accumulation of metals by the albumen gland followed the following order: Fe > Cu > Zn > Al > Mn > Cd > Pb > Ni > Cr > Se > Li > Hg, Al > Fe > Zn > Cu > Mn > Cd > Cr > Pb > Ni > Li > Se >Hg for the digestive organ tissue, Al > Fe > Cu > Zn > Mn > Cr > Pb > Cd > Ni > Se > Li >Hg for the foot tissue, Fe > Al > Zn > Mn > Cu > Cd > Pb > Cr > Ni > Li > Se > Hg for the hepatopancreas, Al > Fe > Zn > Cu > Mn > Cd > Cr > Ni > Li > Se > Pb > Hg for the mantle, Fe > Al > Cu > Zn > Mn > Cd > Cr > Ni > Pb > Se > Li > Hg for the reproductive organ tissue and Al > Fe > Cu > Mn > Zn > Ni > Li > Cr > Pb > Cd > Se for the shells. The conducted Pearson correlation analysis in our study showed statistically significant negative correlations between the Zn, Ni, Fe, Mn, Cr, Al, Li, Pb, Se and Hg concentrations and the examined tissue (Table 5).

In the ground, the concentration of heavy metals follows a similar general pattern as in the snail organisms: Al>Fe>Mn>Zn>Cr> Li> Ni> Pb> Cu> Cd >Hg>Se. Positive correlations were found between the metal concentrations in the ground and the metal concentrations in the examined snail tissues, in particular (Table 6), (1) Nikšić—foot, digestive system, mantle and shell, (2) Malesija—hepatopancreas, reproductive system, foot, digestive system and mantle and (3) Biogradska Gora—reproductive system, foot, digestive system, mantle and shell.

The canonical analysis performed on the metals variability in the examined tissue revealed three distinct clusters (Figure 1b). Most organs such as the reproductive system, albumen gland and foot were clustered together. On the other hand, the hepatopancreas and shell form separate clusters. The first discriminant function explains 40% of the total variance and it is dominated by the Fe content (R = 0.9998). The second discriminant function explains 25% of the total variance and it is dominated by the Zn content (R = 0.9878). The variables with the greatest ability for discrimination along the first discriminant function are Cd, Al and Fe, whereas Se and Ni are those that mostly discriminate snail tissues along the second discriminant function.

The results of our study showed that the shell accumulated the lowest metal amounts. This is in agreement with similar studies conducted on freshwater snails [27,28,29]. Some studies showed that shells play an important role as a sink for some heavy metals [12]. On the other hand, the hepatopancreas accumulates the highest metal content. This is consistent with other studies which revealed that the digestive gland is the tissue which accumulates the highest metal content [2,3,30]. The hepatopancreas and kidney have crucial roles in the detoxification process [12]. As can be seen from Table 4, the accumulation of Zn and Cd in the hepatopancreas are at least 3-fold higher than that in the snail foot. Nowakowaska et al. [12] showed that the hepatopancreatic accumulation of Zn, Cd and Mg was many times higher than that in the kidney and foot. Further, the concentration of Fe and Mn in the hepatopancreas are, respectively, higher that in the foot tissue. In the present study, the Fe concentration of the hepatopancreas was higher by 3.5–4.9-fold in *H. secernenda* and 1.7-fold in *H. vladika* than that in the foot tissue.

### 3.3. Toxic Elements Content in Edible Snails of Montenegro and Comparision with Legal Limit for Food

In our study, the values of toxic elements were compared with those established by the national and EU regulations (EU commission regulation No 1881/2006 of 19 December, 2006) (Table 7). The general mean Pb concentration of the examined snail samples varied from 0.11 ± 0.01 (Malesija) to 0.86 ± 0.1 mg kg^−1^ w.w. (Biogradska Gora), which is below the allowable level for mollusc meat (1.5 mg kg^−1^ w.w.). The mean Hg concentration of the examined snail samples at each site was lower than the legal limit (0.5 mg kg^−1^ w.w.). The general mean Cd concentration of the examined snail tissues was 1.49 ± 0.14 mg kg^−1^ w.w. for the population of *H. secernenda* from Malesija and 1.565 ± 0.0 mg kg^−1^ w.w. for the population of *H. vladika* from Biogradska Gora. These values significantly exceeded the maximum allowable level established by EU regulations (1 mg kg^−1^ w.w.). The obtained values in our study were lower than the values for the *Helix pomatia* populations collected from the wilderness in Moldova, which for Pb ranged between 1.2018 and 48.0840, and between 2.0273 and 3.1281 mg kg^−1^ for Cd [7].

## 4. Conclusions

Our results showed that the concentration of the metals content varied among the snail tissues, which affected their nutritional benefit for human consumption. The hepatopancreas is often considered the tastiest part of the snail meat and, as shown in this study, tends to bioaccumulate selenium and to a lesser extent also zinc. The digestive tract tissue is a target organ for the bioaccumulation of selenium, whose level at the Malesija site reached a concentration of 0.587 mg k^−1^ d.w. Nevertheless, both the hepatopancreas and digestive tract tissues accumulate also toxic elements such as cadmium, whose concentrations in the present study exceeded the permissible limits established by the national and EU regulations. Therefore, the increased content of Cd in our study indicates a possible risk for human health and limits the use of the populations of *Helix vladika* and *H. secernenda* collected from the wilderness as a food source in human nutrition.

## Figures and Tables

**Figure 1 foods-09-00731-f001:**
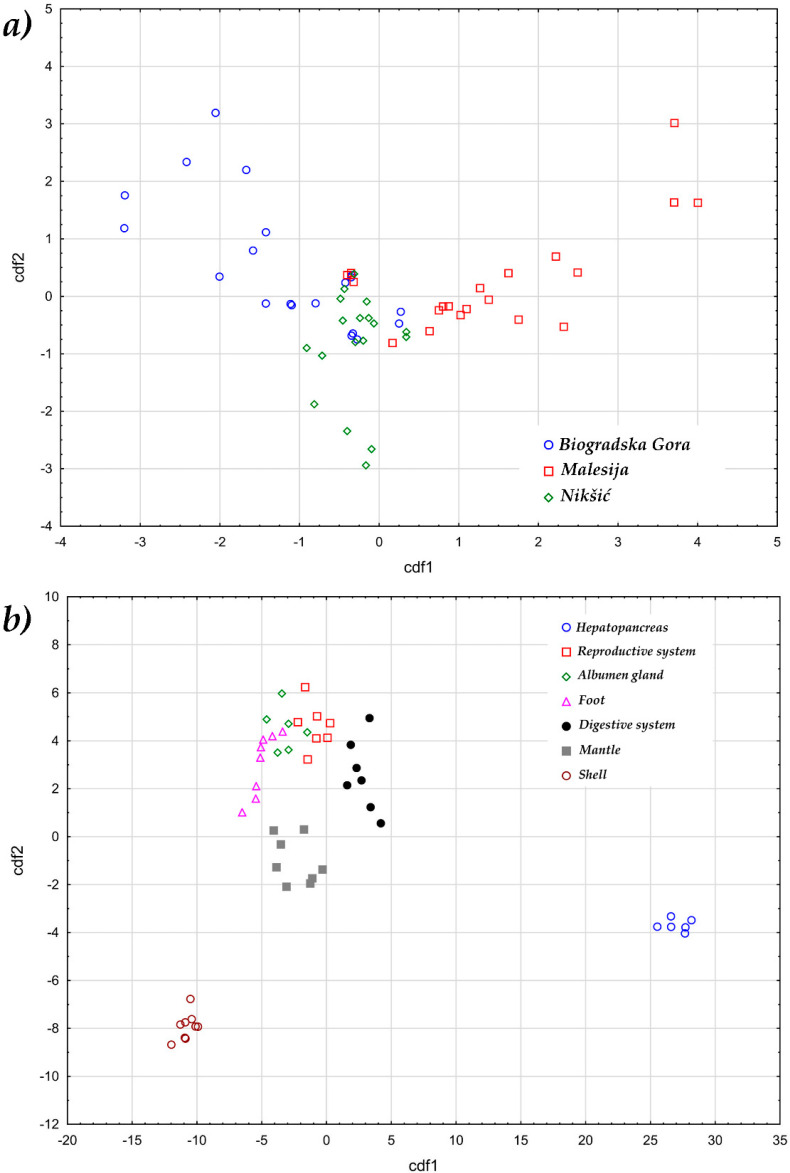
Canonical discriminant score plot of (**a**) examined sampling localities, and (**b**) examined snail tissues.

**Table 1 foods-09-00731-t001:** Mean values (mean ± standard error) of water contents and other allometric parameters of the studied snail population.

Site	Shell Height (cm)	Shell Width (cm)	Wet Weight (g)	Dry Weight (g)	Water Content (%)
Nikšić	4.4 ± 0.4	3.9 ± 0.3	22.0 ± 4.0	9.2 ± 2.1	58.3 ± 8.4
Malesija	4.7 ± 0.2	4.0 ± 0.1	29.8 ± 3.8	10.6 ± 1.6	64.2 ± 5.6
Biogradska Gora	5.3 ± 0.2	4.6 ± 0.5	38.3 ± 5.1	17.4 ± 4.5	55.1 ± 6.4

**Table 2 foods-09-00731-t002:** Operating parameters for the microwave oven (CEM, MDS 2000).

Parameter	Stage				
	1	2	3	4	5
power (%)	100	100	100	100	100
pressure (PSI)	20	40	85	135	175
time (min)	5	10	10	10	5

**Table 3 foods-09-00731-t003:** Recovery (%) rate (mean ± SD) from reference materials MODAS-2 and MODAS-3.

Reference Material	Zn	Ni	Fe	Mn	Cr	Cu	Al	Li	Pb	Cd
Modas 3 (M-3 Her Tis)	99.6 ± 3.1	93.1 ± 3.6	95.2 ± 6.1	101.0 ± 6.9	106.2 ± 7.3	87.5 ± 5.4	-	108.6 ± 3.6	95.9 ± 6.2	93.8 ± 5.9
Modas 2 (M2 BotSed)	113.0 ± 3.3	87.1 ± 2.8	98.2 ± 5.5	107.0 ± 5,1	108.2 ± 3.5	99.4 ± 4.4	87.0 ± 6.5	102.6 ± 3.1	83.5 ± 4.2	97.2 ± 2.9

**Table 4 foods-09-00731-t004:** Minerals and toxic elements content of the different snail tissues from the three sampling sites. The concentrations of elements in the ground and snail tissues were expressed as mean ± standard deviation calculated on a dry weight (d.w.) basis. a,b—the mean values marked with small letters differ statistically significantly at *p* ≤ 0.05 horizontally; A—the mean values marked with capital letters differ statistically significantly at *p* ≤ 0.05 vertically. * Tukey’s *t*-test statistics, ** Kruskal–Wallis statistics. Abbreviations: Gr—ground, Hp—hepatopancreas, Rs—reproductive system, Ag—albumen gland, F—foot, Ds—digestive system, M—mantle, S—shell.

Site	Sample		Zn mg kg^−1^	Ni mg kg^−1^	Fe mg kg^−1^	Mn mg kg^−1^	Cr mg kg^−1^	Cu mg kg^−1^	Al mg kg^−1^	Li mg kg^−1^	Pb mg kg^−1^	Cd mg kg^−1^	Se mg kg^−1^	Hg µg kg^−1^
Nikšić (*Helix secernenda*)	Gr	Mean ± SD	248.5aA * ± 22.7	45.41bbA * ± 1.75	22532.9abA * ± 627.9	1074.9aA * ± 29.6	85.54aA * ± 2.79	42.7 ± 5.1	32212.8A * ± 1483.6	35.08A * ± 1.09	30.92A * ± 0.36	1.24a * ± 0.04	0.06abA * ± 0.01	115.7ab * ± 6.0
	Hp	Mean ± SD	387.0aA * ± 16.9	0.82b * ± 0.12	205.7ab * ± 8.7	405.1aA * ± 11.0	3.80a * ± 0.19	45.7 ± 5.9	214.9 ± 3.5	0.18 ± 0.04	1.96 ± 0.39	1.43a * ± 0.03	0.19ab * ± 0.01	20.3a * ± 0.8
		BCF	1.56 ± 0.74	0.02 ± 0.07	0.01 ± 0.01	0.38 ± 0.37	0.04 ± 0.07	1.07 ± 1.16	0.00 ± 0.00	0.00 ± 0.04	0.06 ± 1.08	1.15 ± 0.75	3.35 ± 0.64	0.17 ± 0.13
	Rs	Mean ± SD	41.4a ± 2.5	0.71b ± 0.33	69.4 ± 12.6	14.2 ± 1.2	0.86 ± 0.12	98.4A ± 4.3	62.0 ± 1.9	0.04 ± 0.00	0.90 ± 0.12	1.10 ± 0.15	0.18 ± 0.01	3.7 ± 0.3
		BCF	0.17 ± 0.11	0.02 ± 0.19	0.00 ± 0.02	0.01 ± 0.04	0.01 ± 0.04	2.30 ± 0.84	0.00 ± 0.00	0.00 ± 0.00	0.03 ± 0.33	0.89 ± 3.75	3.09 ± 1.36	0.03 ± 0.05
	Ag	Mean ± SD	33.2aA * ± 3.5	1.21a * ± 0.03	26.2a * ± 5.0	17.7a * ± 1.5	0.36a * ± 0.05	70.4 ± 11.5	8.6 ± 1.2	0.12a * ± 0.00	0.25a * ± 0.06	0.57 ± 0.08	0.11 ± 0.00	4.0 ± 0.00
		BCF	0.13 ± 0.15	0.03 ± 0.02	0.00 ± 0.01	0.02 ± 0.05	0.00 ± 0.02	1.64 ± 2.25	0.00 ± 0.00	0.00 ± 0.00	0.01 ± 0.17	0.46 ± 2	1.88 ± 0.45	0.03 ± 0.00
	F	Mean ± SD	39.5 ± 2.4	0.66a * ± 0.16	41.7a * ± 4.6	7.6a * ± 0.2	1.29 ± 0.10	53.6a * ± 2.5	138.4 ± 29.1	0.09a * ± 0.05	0.21 ± 0.13	0.48 ± 0.14	0.11 ± 0.00	5.7 ± 0.1
		BCF	0.16 ± 0.10	0.01 ± 0.09	0.00 ± 0.01	0.01 ± 0.01	0.01 ± 0.03	1.25 ± 0.49	0.00 ± 0.02	0.00 ± 0.04	0.01 ± 0.36	0.38 ± 3.5	1.95 ± 0.27	0.05 ± 0.02
	Ds	Mean ± SD	63.6 ± 5.0	0.64 ± 0.17	181.3 ± 13.2	14.7 ± 0.5	2.71 ± 0.42	68.3 ± 16.4	81.2 ± 11.2	0.31 ± 0.27	0.94a ** ± 0.04	4.81a ** ± 1.03	0.19 ± 0.01	9.2a ** ± 2.0
		BCF	0.25 ± 0.22	0.01 ± 0.10	0.01 ± 0.02	0.01 ± 0.02	0.03 ± 0.15	1.6 ± 3.21	0.00 ± 0.01	0.01 ± 0.25	0.03 ± 0.11	3.88 ± 25.75	3.26 ± 1.27	0.08 ± 0.33
	M	Mean ± SD	78.2 ± 2.8	0.13 ± 0.03	91.8 ± 25.9	49.6 ± 5.6	0.82 ± 0.26	51.4 ± 0.2	116.7 ± 0.4	0.31 ± 0.06	0.01 ± 0.00	2.72 ± 0.09	0.10 ± 0.00	5.4 ± 0. 3
		BCF	0.31 ± 0.12	0.00 ± 0.02	0.00 ± 0.04	0.05 ± 0.19	0.01 ± 0.09	1.20 ± 0.04	0.00 ± 0.00	0.01 ± 0.05	0.00 ± 0.00	2.19 ± 2.25	1.79 ± 0.54	0.05 ± 0.05
	S	Mean ± SD	3.7aA * ± 0.1	0.96a * ± 0.17	26.8a * ± 4.7	5.7 ± 0.7	0.50a * ± 0.17	12.9A * ± 9.4	502.9 ± 29.6	0.41 ± 0.01	0.12 ± 0.11	0.08aA * ± 0.02	0.01aA * ± 0.00	0.8a * ± 0.00
		BCF	0.01 ± 0.00	0.02 ± 0.10	0.00 ± 0.00	0.00 ± 0.02	0.01 ± 0.06	0.30 ± 1.84	0.01 ± 0.02	0.01 ± 0.01	0.00 ± 0.30	0.06 ± 0.5	0.24 ± 0.18	0.00 ± 0.00
Malesija (*Helix secernenda*)	Gr	Mean ± SD	108.8aA * ± 7.9	59.04A ± 5.41	27068.7aA * ± 1977.4	758.4A * ± 18.0	110.56A * ± 11.42	27.3a * ± 0.9	50288.1aA * ± 3197.3	51.47aA * ± 4.60	28.32aA * ± 1.29	0.65 ± 0.02	0.06aA * ± 0.01	121 ± 11
	Hp	Mean ± SD	253.4A * ± 18.5	2.12 ± 0.36	463.9 ± 23.8	32.4 ± 0.3	3.41 ± 0.58	40.4 ± 0.5	484.0 ± 4.5	0.72 ± 0.29	0.99 ± 0.29	16.38A * ± 1.46	0.27A * ± 0.02	24 ± 0.4
		BCF	2.33 ± 2.34	0.03 ± 0.07	0.02 ± 0.01	0.04 ± 0.02	0.03 ± 0.05	1.48 ± 0.55	0.01 ± 0.00	0.01 ± 0.06	0.03 ± 0.22	25.2 ± 73	4.44 ± 2.22	0.19 ± 0.04
	Rs	Mean ± SD	61.3a * ± 1.1	0.60b * ± 0.32	89.5ab * ± 6.0	10.4ab * ± 0.9	0.92ab * ± 0.01	67.1a * ± 4.2	96.0ab * ± 8.1	0.04 ± 0.00	0.07a * ± 0.04	2.99aA * ± 0.24	0.24abA * ± 0.02	5 ± 0.3
		BCF	0.56 ± 0.14	0.01 ± 0.06	0.00 ± 0.00	0.01 ± 0.05	0.01 ± 0.00	2.45 ± 4.67	0.00 ± 0.00	0.00 ± 0.00	0.00 ± 0.03	4.6 ± 12	3.98 ± 2.67	0.04 ± 0.03
	Ag	Mean ± SD	45.7a * ± 1.6	0.24a * ± 0.32	99.5a * ± 10.5	10.9a * ± 0.7	0.63a * ± 0.21	41.1 ± 2.7	42.0 ± 11.8	0.08 ± 0.06	0.08a * ± 0.02	1.69A * ± 0.08	0.24abA * ± 0.01	3.5 ± 0.0
		BCF	0.42 ± 0.20	0.00 ± 0.06	0.00 ± 0.00	0.01 ± 0.04	0.00 ± 0.02	1.50 ± 3	0.00 ± 0.00	0.00 ± 0.01	0.00 ± 0.01	2.6 ± 4	3.93 ± 1.55	0.03 ± 0.00
	F	Mean ± SD	56.9 ± 0.4	0.28a * ± 0.06	134.3a * ± 15.4	8.9a * ± 0.1	1.18a * ± 0.44	75.8 ± 15.2	130.9 ± 4.2	0.12 ± 0.00	0.18a * ± 0.03	0.91 ± 0.09	0.18 ± 0.01	6.7 ± 0.2
		BCF	0.52 ± 0.05	0.00 ± 0.01	0.00 ± 0.01	0.01 ± 0.00	0.01 ± 0.04	2.77 ± 16.9	0.00 ± 0.00	0.00 ± 0.00	0.01 ± 0.02	1.4 ± 4.5	2.96 ± 1.22	0.05 ± 0.01
	Ds	Mean ± SD	41.1a * ± 2.5	1.44a * ± 0.13	805.0a * ± 67.6	26.6a * ± 2.1	4.84 ± 0.43	36.9 ± 0.7	1411.1a * ± 172.0	1.26a * ± 0.07	0.42 ± 0.06	5.24 ± 0.51	0.59A ± 0.01	13.3 ± 0. 4
		BCF	0.38 ± 0.32	0.02 ± 0.02	0.03 ± 0.03	0.03 ± 0.12	0.04 ± 0.04	1.35 ± 0.77	0.03 ± 0.05	0.02 ± 0.01	0.01 ± 0.05	8.06 ± 25.5	9.62 ± 1.67	0.11 ± 0.04
	M	Mean ± SD	62.4 ± 12.7	0.42a * ± 0.25	224.8 ± 14.5	29.0 ± 5.4	1.24a * ± 0.10	64.3 ± 3.9	432.2 ± 28.9	0.28a * ± 0.00	0.21 ± 0.04	2.16 ± 0.75	0.17 ± 0.02	6.6 ± 0.6
		BCF	0.57 ± 1.60	0.01 ± 0.05	0.01 ± 0.01	0.04 ± 0.3	0.01 ± 0.01	2.35 ± 4.33	0.01 ± 0.01	0.00 ± 0.00	0.01 ± 0.03	3.32 ± 37.5	2.83 ± 2	0.05 ± 0.05
	S	Mean ± SD	3.7aA ** ± 0.2	0.95a ** ± 0.44	25.1a ** ± 10.4	2.3a ** ± 1.0	0.45a ** ± 0.09	9.0aA ** ± 4.6	512.4a ** ± 46.8	0.43a ** ± 0.05	0.16a ** ± 0.09	0.10aA ** ± 0.02	0.01a ** ± 0.00	1.9A ** ± 0.1
		BCF	0.03 ± 0.02	0.02 ± 0.08	0.00 ± 0.00	0.00 ± 0.05	0.00 ± 0.01	0.33 ± 5.11	0.01 ± 0.01	0.01 ± 0.01	0.00 ± 0.07	0.15 ± 1	0.19 ± 0.44	0.01 ± 0.01
Biogradska Gora (*Helix vladika*)	Gr	Mean ± SD	153.7A * ± 15.4	41.57A * ± 6.68	29053.7A ** ± 1499.1	2895.7A ± 172.1	87.63A * ± 1.59	36.7 ± 1.9	37653.1A * ± 3416.4	96.38A * ± 16.49	49.72A * ± 5.53	0.99 ± 0.04	0.05 ± 0.01	260.0 ± 1.1
	Hp	Mean ± SD	151.6A * ± 6.9	0.95a * ± 0.07	149.3 ± 16.4	350.6A * ± 1.9	0.95 ± 0.16	33.2 ± 11.5	100.1 ± 2.6	0.10 ± 0.00	5.46 ± 0.18	11.28 ± 0.17	0.11 ± 0.02	53.9A * ± 2.1
		BCF	0.99 ± 0.44	0.02 ± 0.01	0.00 ± 0.01	0.12 ± 0.01	0.01 ± 0.10	0.90 ± 6.05	0.00 ± 0.00	0.00 ± 0.00	0.11 ± 0.03	11.39 ± 4.25	2.35 ± 2.29	0.21 ± 1.91
	Rs	Mean ± SD	61.0 ± 4.4	1.28 ± 0.35	87.7 ± 3.2	13.6 ± 0.5	1.20 ± 0.28	53.9 ± 6.5	68.4 ± 10.5	0.08 ± 0.00	0.12 ± 0.00	2.00 ± 0.07	0.13 ± 0.01	6.5A * ± 0.5
		BCF	0.39 ± 0.28	0.03 ± 0.05	0.00 ± 0.00	0.00 ± 0.00	0.01 ± 0.18	1.47 ± 3.42	0.00 ± 0.00	0.00 ± 0.00	0.00 ± 0.00	2.02 ± 1.75	2.73 ± 1.71	0.02 ± 0.45
	Ag	Mean ± SD	51.7 ± 1.4	0.91 ± 0.28	64.0 ± 0.3	23.8 ± 0.1	0.51 ± 0.33	53.5 ± 1.6	16.6 ± 0.6	0.08 ± 0.00	2.78 ± 1.13	1.66 ± 0.26	0.10 ± 0.00	5.3 ± 0.3
		BCF	0.33 ± 0.09	0.02 ± 0.04	0.00 ± 0.00	0.01 ± 0.00	0.01 ± 0.21	1.46 ± 0.84	0.00 ± 0.00	0.00 ± 0.00	0.05 ± 0.20	1.68 ± 6.5	2.1 ± 0.71	0.02 ± 0.27
	F	Mean ± SD	52.9a * ± 1.8	0.52b * ± 0.11	87.1 ± 7.7	16.0 ± 1.1	1.32ab * ± 0.26	49.0 ± 1.3	106.5 ± 12.0	0.03ab * ± 0.05	1.76 ± 0.36	0.65 ± 0.04	0.11 ± 0.01	5.4 ± 0.2
		BCF	0.34 ± 0.11	0.01 ± 0.01	0.00 ± 0.00	0.00 ± 0.01	0.01 ± 0.16	1.33 ± 0.68	0.00 ± 0.00	0.00 ± 0.00	0.03 ± 0.06	0.65 ± 1	2.2 ± 1.28	0.02 ± 0.18
	Ds	Mean ± SD	63.7a * ± 2.7	0.20a * ± 0.10	111.9a * ± 11.1	48.7a * ± 5.9	1.15a * ± 0.58	42.4 ± 3.5	73.6 ± 8.0	0.03a * ± 0.05	3.09a * ± 0.03	6.10 ± 0.86	0.11 ± 0.02	42.1 ± 33.0
		BCF	0.41 ± 0.17	0.00 ± 0.01	0.00 ± 0.01	0.02 ± 0.03	0.01 ± 0.36	1.15 ± 1.84	0.00 ± 0.00	0.00 ± 0.00	0.06 ± 0.00	6.16 ± 21.5	2.16 ± 2.85	0.16 ± 30
	M	Mean ± SD	44.5 ± 2.0	0.22 ± 0.11	86.8 ± 3.1	14.9 ± 2.4	1.00 ± 0.15	68.3 ± 3.6	207.6 ± 21.6	0.11 ± 0.12	0.06 ± 0.00	2.61 ± 0.11	0.08 ± 0.01	6.8 ± 0.1
		BCF	0.29 ± 0.13	0.00 ± 0.02	0.00 ± 0.00	0.00 ± 0.01	0.01 ± 0.10	1.86 ± 1.89	0.00 ± 0.01	0.00 ± 0.01	0.00 ± 0.00	2.63 ± 2.75	1.55 ± 2.14	0.03 ± 0.09
	S	Mean ± SD	4.2A ** ± 0.3	1.15a ** ± 0.24	2.8a ** ± 0.8	6.9a ** ± 1.3	0.26 ± 0.15	7.8A ** ± 1.8	445.6 ± 17.2	0.44 ± 0.12	0.16 ± 0.04	0.03aA ** ± 0.01	0.01A ** ± 0.00	1.5 ± 0.0
		BCF	0.03 ± 0.02	0.03 ± 0.03	0.00 ± 0.00	0.00 ± 0.01	0.00 ± 0.10	0.21 ± 1.95	0.01 ± 0.00	0.00 ± 0.00	0.00 ± 0.00	0.03 ± 0.25	0.14 ± 0.28	0.01 ± 0.00

**Table 5 foods-09-00731-t005:** Pearson correlations between the metal concentrations and snail tissue and sampling localities, respectively. * *p* ≤ 0.05.

Variable	Zn	Ni	Fe	Mn	Cr	Cu	Al	Li	Pb	Cd	Se	Hg
Tissue	−0.662 *	−0.586 *	−0.590 *	−0.541 *	−0.485 *	−0.222	−0.572 *	−0.536 *	−0.598 *	−0.244	−0.344 *	−0.591 *
Locality	−0.167	−0.027	0.016	0.120	−0.084	−0.181	0.001	0.107	0.092	0.177	−0.114	−0.031

**Table 6 foods-09-00731-t006:** Pearson correlation between the selected organs and sampling sites. Abbreviations: Gr—ground, Hp—hepatopancreas, Rs—reproductive system, Ag—albumen gland, F—foot, Ds—digestive system, M—mantle, S—shell. * *p* ≤ 0.05.

Site	Hp	Rs	Ag	F	Ds	M	S
Nikšić	0.233	0.365	0.001	0.798 *	0.865 *	0.865 *	0.757 *
Malesija	0.885 *	0.739 *	0.555	0.859 *	0.999 *	0.891 *	0.926 *
Biogradska Gora	0.365	0.821*	0.395	0.950*	0.736 *	0.902 *	0.733 *

**Table 7 foods-09-00731-t007:** Mean concentrations of metals in the studied snails from the three sampling sites of Montenegro and the legal limits (guidelines EU No 1881/2006), referred to as wet weight.

Metals	Investigated Sites			Legal Limit (1881/2006/EU)
	Nikšić	Malesija	Biogradska Gora	
Pb (mg kg^−1^)	0.26 ± 0.00	0.11 ± 0.01	0.86 ± 0.12	1.5 (molluscs)
Cd (mg kg^−1^)	0.67 ± 0.05	1.49 ± 0.14	1.56 ± 0.09	1.0 (molluscs)
Hg (µg kg^−1^)	2.94 ± 0.00	3.6 ± 0.0	9.00 ± 0.00	500 (fish)

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
