# Peer review of "Toxic Elements and Mineral Content of Different Tissues of Endemic Edible Snails (Helix vladika and H. secernenda) of Montenegro"

_foods, 2020, doi:10.3390/foods9060731_

Round 1
Reviewer 1 Report
The work reports a study on mineral content of different tissues od endemic edible snails collected in Montenegro. The topic is interesting, but paper did not provide enough description of the food samples, and there are some important mistake concerning the comparison of metals concentration (in mg kg-1 dry weight) with EU legal limits (in mg kg-1 wet weight). The paper fit with aims and scope of the journal. Major revisions are required in accordance with the following advices.
General advices
- Pay attention to the significant figures. The general rule is that experimental uncertainty (Standard Deviation) should be rounded to one significant digit. In Tables and along the text authors reported in some cases too many digits. Please correct.
- The manuscript should be revised in depth for English language.
Title
I suggest adding in the title and key words “Toxic elements”
Introduction
Line 51. “meat a”
Line 74. Authors should explain the reason they studied, beside mineral composition of essential elements, the specific toxic elements Cd, Pb , Ni, Cr, Hg, and so on. Are they particularly toxic for humans? (the answer is obvious, but authors should explain briefly the reason)
Material and methods
Lines 83-86. Which species did you collected in each site? How may specimens for each species from each site?
Table 1. Too many digits.
Chemical analysis: authors should indicate the calibration method used
Line 104. To constant weight.
Line 109. Samples were weighed
Line 111. Was added
Lines117-118. Reference for LOD calculation
Line 118. Delete ug/l after Al
Line 120: grammatical mistakes
Table 3: no standard deviation? Did authors perform only 1 analysis?
Line 127. Total Hg (THg)
Line 133-135. Why authors performed the recovery test for Hg only in one reference material?
Results and Discussion
Table 4. Se is not a metal. Please indicate the measure unit: mg kg-1 dry weight or wet weight? Given the premises, reader expects to find separate results between the two indagated species.
Line 181-183. In my opinion, author should put this sentence at the beginning of the result section, to justify why in Table 4 authors showed the metal conc considering together the two indagated species.
Line 183-184. was the higher content of Se and Li observed for all tissues? The higher concentration in one species is not enough to assert that this species has a higher bioaccumulation ability. The bioaccumulation refers to the ratio between metal content in the body and metal content in the substrate. I think that authors should explain better this part. The same along all the text. See for example references reported below, and: Cristina Truzzi, Anna Annibaldi, Federico Girolametti, Leonardo Giovannini, Paola Riolo, Sara Ruschioni, Ike Olivotto, Silvia Illuminati. A Chemically Safe Way to Produce Insect Biomass for Possible Application in Feed and Food Production. International Journal of Environmental Research and Public Health. Special issue 2020, 17, 2121; doi:10.3390/ijerph17062121
Lines 185-196. About Se concentration, it would be interesting to perform a comparison with other kind of food. For example, two recent interesting papers reported concentration of Se in Mediterranean tuna e in new food such as insects (T. molitor). Moreover, they studied the health benefit value relating Hg content to Se content.
- Anna Annibaldi, Cristina Truzzi, Oliana Carnevali, Paolo Pignalosa, Martina Api, Giuseppe Scarponi and Silvia Illuminati. Determination of Hg in farmed and wild Atlantic Bluefin Tuna (Thunnus thynnus L.) muscle. Molecules special issue, 2019, 24 1273.
- Cristina Truzzi, Silvia Illuminati, Federico Girolametti, Matteo Antonucci, Giuseppe Scarponi, Sara Ruschioni, Paola Riolo, Anna Annibaldi. Influence of feeding substrates on the presence of toxic metals (Hg, Cd, Pb, Ni, As) in larvae of Tenebrio molitor: risk assessment for human consumption. International Journal of Environmental Research and Public Health. Special issue 2019, 16, 4815; https://doi:10.3390/ijerph16234815
Line 198. Please explain which EU regulations authors refer. Be careful that legal limits are reported in mg kg-1 wet weight, whereas your data are in d.w. You must transform data in w.w for comparison with legal limit.
Line 199. d.m. or d.w. (line 186)? be consistent. Authors must calculate metal content in wet weigh, to compare concentrations with legal limits.
Line 206. 3.09±4.03 is not lower than 2.0273-3.1281.
Line 208. authors should explain in Material and Method where each species were collected
Line 216-217. Canonical analysis discriminates also between Biogradska Gora and Niksic on cdf2.
Authors should underline that Hg content is lower than legal limit in all samples
Lines237-238. Other elements also showed significant differences between the ground and examined tissues.
Line 277-288. It is more correct to compare the Bioaccumulation factor, instead of the only absolute concentration of elements.
Canonical Discriminant Analysis. Authors should report also the loading scores, to understand which variable affect the variability of data. For example, in Fig 1b, which are the variables that discriminate between shell and hepatopancreas along the cdf1? And which are the variables that discriminate between shell, hepatopancreas and other tissues along the cdf2. The same for Fig1a.
Author Response
"Please see the attachment."

Reviewer 2 Report
The article provide the contents of zinc (Zn), manganese (Mn), copper (Cu), aluminum (Al), cadmium (Cd), lead (Pb), nickel (Ni), iron (Fe), chromium (Cr), lithium (Li), selenium (Se) and mercury (Hg) in different snail tissue of the two most abundant edible snails of genus Helix in Montenegro.
General comments
- In the line 188 the bibliography is in two different ways, by numbers and by author’s name, please review all bibliographic references. “These values were higher in comparison with those reported for raw meat of Roman snails from Poland (0.09 mg kg) [2] but lower in comparison with those reported for the meat of H. pomatia snails from Moldavia, Ukraine and Russia which varied in the range between 0.130 and 0.423 mg kg-1 189 (Toader-Williams & Golubkina 2009).
- The authors could put a map and sign the collection samples localization.
- In my opinion the table 4 it should be represented on a graph or on a graph panel, so that the results are more visual, so they are difficult to analyze.
- In figure 1B we can see that the Hepatopancreas it is the one with the highest bioaccumulation, being the Shell the one with the least bioaccumulation. However, in the conclusions it is mentioned that hepatopancreas and digestive tract tissues are the main snail tissues to accumulate toxic elements such as cadmium and lead whose concentrations in the present study exceed the permissible limits. Pleased explain?
- The conclusions should be improved, and answer the starting question.
Author Response
"Please see the attachment."

Reviewer 3 Report
Vukasinovic- Pesic et al studied the Mineral content of different tissues of endemic edible snails (Helix vladika and H. secernenda) of Montenegro found interesting but additional research would be of benefit to better understand the possible health risk imposed by consuming edible snails from Montegero.
Major comments:
I recommend the authors to study the bioavailability of lead and cadmium from snail samples using in vitro biomimetic system. For acquiring this data, author can either perform in vitro biomimetic assay in Caco2 cells (possessing all gastric juices in vitro) or perform real-time PCR experiments to understand how the heavy metal transporters (p-gp, MDR, MRP etc) function during lead and cadmium toxicity. I recommend using Caco2 cell lines for this study.
Minor comments:
1. English must be improved
2. Please re-check the references according to the journal format
3. Please use abbreviations when you first write the word. For eg: abbreviate Zn in the line 68 instead of line 74. Likewise please check all abbreviations
4. Please write the full form of M, N and B inside the Table 4, or write the full form of M, N, and B in the table legend. Like Niksic (N)
5. Line 232-240 which site? N, M or B?
Author Response
"Please see the attachment."

Reviewer 4 Report
Recommendation: Include selected data on the content of substances in the abstract (for example: Hepatopancreas accumulates the highest concentrations of Zn, Ni, Mn, Pb, Cd and Hg...)
Has it been stated that the work is in accordance with the Ethical Guidelines for the Use of Animals in Research?
...Any experimental work must also have been conducted in accordance with relevant national legislation on the use of animals for research. For further guidance authors should refer to the Code of Practice for the Housing and Care of Animals Used in Scientific Procedures.
Author Response
"Please see the attachment."

Round 2
Reviewer 1 Report
General comment
Authors should take time to carefully revise the paper. In the modified text (and in the new added text) there are a lot of grammatical mistakes, the results and discussion are difficult to read, the tables and the text still contain a lot of numbers with too many digits, and some answer to my comments are incomplete.
My comments were added in “Response to Reviews” and were shown in red
RESPONSE TO REVIEWS
The authors would like to thank anonymous reviewers for correcting our manuscript by contributing with valuable suggestions and comments.
In revised version we followed all referee suggestions and proposal for corrections.
Furthermore, we tried carefully to correct point by point each comment and give the appropriate explanation to those specific requests that have not been changed. Lines in our answers to referee comments refer to the “Clean Version” of revised version is edited for English, and carefully checked for typos, double spaces and so on.
Reviewers' comments:
Rev. #1
Title. I suggest adding in the title and key words “Toxic elements”
We added “toxic elements” into Key words
Authors forgot to insert “toxic elements” in the title. Toxic elements cannot named “minerals”, but authors must specify that they studied mineral elements and toxic elements.
Introduction
Line 51. “meat a”
Changed. ok
Line 74. Authors should explain the reason they studied, beside mineral composition of essential elements, the specific toxic elements Cd, Pb, Ni, Cr, Hg, and so on. Are they particularly toxic for humans? (the answer is obvious, but authors should briefly explain the reason)
Thank you for this suggestion, we have done so. Lines 74-77 “The obtained data are allowed to identify different snail tissues as a potential source of some essential elements for human nutrition on one side, and on another one to evaluate the risk assessment of accumulation of toxic elements for the human consumption.”
Authors did not answer to my comment: they must explain why they indagated these specific elements: are they priority pollutants? Are they dangerous? See for example the introduction of the following paper: Cristina Truzzi, et al. A Chemically Safe Way to Produce Insect Biomass for Possible Application in Feed and Food Production. International Journal of Environmental Research and Public Health. Special issue 2020, 17, 2121; doi:10.3390/ijerph17062121
Material and methods
Lines 83-86. Which species did you collected in each site? How may specimens for each species from each site?
Explained. Lines 83-89 “The Biogradska Gora site where H. vladika was collected is located in a protected area, the Biogradska Gora National Park. Two other sites, Malesija (Skorać) and Nikšić (Dragovoljići) where inhabited by H. secernenda and are located outside the motorized traffic and were not affected to any source of pollution. Samples were taken in May (at Biogradska Gora and Nikšić) and in September 2018 (Malesija). From the each site 30 snail specimens with almost a similar size were randomly taken.” ok
Line 92: where population
Table 1. Too many digits.
Changed.
22.0±4.0 must change in 22±4, 29.8±3.8 must change in 29±4, and so on for all numbers with 2 digits before the comma. The general rule is that experimental uncertainty (Standard Deviation) should be rounded to one significant digit
Chemical analysis: authors should indicate the calibration method used
Calibration curve technique was used for the quantification of toxic elements. Single-element calibration standards (Merck, Germany) were used as the stock standards for preparing the working standards. Working standards were prepared by serial volume/volume dilution in glass volumetric flasks with 0.2% HNO3 (Pb, Cd) or 15% HNO3 (others elements) in MilliQ-water (name of deionizer, firm, city, country). The linearity was considered satisfactory and acceptable if the correlation coefficient, r exceeded 0.999 and in case of Pb and Cd we found correlation coefficient, r2≥0.9997. For quality control, blanks, calibration standards of each analyte and standard reference materials were tested during each batch.
Authors must put this test in the text.
Line 104. To constant weight.
Changed.
Line 110. Replace “dry weights” with “dry weight”
Line 109. Samples were weighed
Changed. ok
Line 111. Was added
Changed. ok
Lines117-118. Reference for LOD calculation
Included – Lines 113-114. The limits of detection (LOD) were determined and calculated based on the data from the calibration curve according Wenzl et al. 2016. Wenzl, T., Haedrich, J., Schaechtele, A., Robouch, P., Stroka, J., Guidance Document on the Estimation of LOD and LOQ for Measurements in the Field of Contaminants in Feed and Food; EUR 28099, Publications Office of the European Union, Luxembourg, 2016, ISBN 978-92-79-61768-3; doi:10.2787/8931. ok
Line 123. “The limits of detection were determined”” or “the limit of detection was determined”
Line 118. Delete ug/l after Al
Done
Line 125. It would be better to write “The calculated LODs were as follows (µg/l): Al 50, Fe 4.0, …….
Line 120: grammatical mistakes
All grammatical mistakes fixed. ok
Table 3: no standard deviation? Did authors perform only 1 analysis?
Standard deviations have been inserted in Table 3 as ± value.
Title: Recovery (%) rate (mean±SD) from…..
The footnotes (Line 134) should be deleted. In the table, below the name of CRM in the cell, authors should write the number of replicates (n=..)
There are too many digits. e.g. In 99.6±3.1, the SD contains 2 digits. The general rule is that experimental uncertainty (Standard Deviation) should be rounded to one significant digit, and the mean must be stopped to the same decimal place of the SD. So, 99.6±3.1 should be written as 100±3. The same for all numbers reported in Table 3.
Line 127. Total Hg (THg)
Changed. Line 124. ok
Line 133-135. Why authors performed the recovery test for Hg only in one reference material?
We check it with two additional reference materials but they expired, that is why we didn’t give the data, though the recovery tests for Hg were acceptable. For future experiments we will share reference material between laboratories.
The Reference Materials MODAS-2 and MODAS-3 that authors used for other elements can be used for Hg?
Results and Discussion
Table 4. Se is not a metal. Please indicate the measure unit: mg kg-1 dry weight or wet weight? Given the premises, reader expects to find separate results between the two indagated species.
We acknowledge the referee suggestion. We changed title of a table and gave measure unit for each of elements. In addition, we have listed the names of the species studied at each site. See Table 4.
I suggest writing in the title of Table 4 “minerals and toxic elements content…”. Moreover, replace “contents” with “content”.
The sentences in the title “the mean concentration of elements in all snail tissues (St) for each site……..The bioaccumulation factor….” should be moved to footnotes.
Be carful to insert all data in this table. It became a bit confusing. I suggest adding another table where author should put data of ST in wet weight, so the reader is facilitated to find data during the discussion. Moreover, it is not necessary to report data in w.w. for all studied elements, but only for toxic elements that have a legal limit (reported in w.w).
Again, there are too many digits. The general rule is that experimental uncertainty (Standard Deviation) should be rounded to one significant digit, and the mean must be stopped to the same decimal place of the SD.
Line 181-183. In my opinion, author should put this sentence at the beginning of the result section, to justify why in Table 4 authors showed the metal conc considering together the two indagated species.
We followed this referee suggestion. Lines 169-183
The sentence of line 187-188 should be moved after the sentence of lines 189-190.
Lines 189-90- The mean concentrations of analysed minerals and toxic elements in the tissues of studied snail populations from three sampling sites in Montenegro, together with Bioaccumulation Factor (BAF), were reported in Table 4.
Line 183-184. was the higher content of Se and Li observed for all tissues? The higher concentration in one species is not enough to assert that this species has a higher bioaccumulation ability. The bioaccumulation refers to the ratio between metal content in the body and metal content in the substrate. I think that authors should explain better this part. The same along all the text. See for example references reported below, and: Cristina Truzzi, Anna Annibaldi, Federico Girolametti, Leonardo Giovannini, Paola Riolo, Sara Ruschioni, Ike Olivotto, Silvia Illuminati. A Chemically Safe Way to Produce Insect Biomass for Possible Application in Feed and Food Production. International Journal of Environmental Research and Public Health. Special issue 2020, 17, 2121; doi:10.3390/ijerph17062121
The reviewer is right. Following this suggestion, we gave the bioaccumulation factor (BAF) for each element / tissue (Table 4). The bioaccumulation factor (BAF) was calculated as BAF = concentration of the target element in the snail tissue / concentration of the same element in the ground (Lines 149-150). In revised draft we used BAF to discuss bioaccumulation abilities of specific tissues (e.g.,lines 184-186, 251-252, 265-268, 276-277, 279-280, 283). ok
Lines 185-196. About Se concentration, it would be interesting to perform a comparison with other kind of food. For example, two recent interesting papers reported concentration of Se in Mediterranean tuna e in new food such as insects (T. molitor). Moreover, they studied the health benefit value relating Hg content to Se content.
- Anna Annibaldi, Cristina Truzzi, Oliana Carnevali, Paolo Pignalosa, Martina Api, Giuseppe Scarponi and Silvia Illuminati. Determination of Hg in farmed and wild Atlantic Bluefin Tuna (Thunnus thynnus L.) muscle. Molecules special issue, 2019, 24 1273.
- Cristina Truzzi, Silvia Illuminati, Federico Girolametti, Matteo Antonucci, Giuseppe Scarponi, Sara Ruschioni, Paola Riolo, Anna Annibaldi. Influence of feeding substrates on the presence of toxic metals (Hg, Cd, Pb, Ni, As) in larvae of Tenebrio molitor: risk assessment for human consumption. International Journal of Environmental Research and Public Health. Special issue 2019, 16, 4815; https://doi:10.3390/ijerph16234815
Yes, the referee is right, and we acknowledge this suggestion. As a results we modified part dealing with Se results and compared our results with other kind of foods (Lines 195-207). The both reference, i.e., Annibaldi et al. 2019, and Truzzi et al. 2019 were found useful for comparison and they are discussed and cited. ok
Line 198. Please explain which EU regulations authors refer. Be careful that legal limits are reported in mg kg-1 wet weight, whereas your data are in d.w. You must transform data in w.w for comparison with legal limit.
The referee is right. In Table 4 we gave values for the mean concentrations of studied elements in a snail samples per each site on a basis of a dry weight and wet weight allowing comparison with legal limit.
Authors did not report the EU regulation they refer.
Line 199. d.m. or d.w. (line 186)? be consistent. Authors must calculate metal content in wet weigh, to compare concentrations with legal limits.
We used d.w. (dry weight) throughout the text. See previous comment. ok
Line 206. 3.09±4.03 is not lower than 2.0273-3.1281.
Changed. Lines 215-217 ok
Line 208. authors should explain in Material and Method where each species were collected
Explained. See Lines 83-89 ok
Line 216-217. Canonical analysis discriminates also between Biogradska Gora and Niksic on cdf2.
The resulting the scatter plot of the first and second canonical scores shows partly overlap of Biogradska Gora and Nikšić localities suggesting that these two sites share similarities in metal content variability (Lines 228-230). ok
Authors should underline that Hg content is lower than legal limit in all samples
Undelined. Lines 211-212. ok
Lines237-238. Other elements also showed significant differences between the ground and examined tissues.
Thank you for this suggestion, we have done so. Lines 246-248. ok
Line 277-288. It is more correct to compare the Bioaccumulation factor, instead of the only absolute concentration of elements.
The referee is right. In Table 4 we gave values for bioaccumulation factor for each element / tissue. ok
Canonical Discriminant Analysis. Authors should report also the loading scores, to understand which variable affect the variability of data. For example, in Fig 1b, which are the variables that discriminate between shell and hepatopancreas along the cdf1? And which are the variables that discriminate between shell, hepatopancreas and other tissues along the cdf2. The same for Fig1a.
Changed. For Fig1a: “The variable with the greatest ability for discrimination along the first discriminant was Fe, while Zn play important roles in the discrimination along the second discriminant function.”(Lines 233-234). For Fig 1b “The variables with the greatest ability for discrimination along the first discriminant function are Cd, Al, and Fe, whereas Se and Ni are those that mostly discriminate snail tissues along the second discriminant function.” (Lines 334-336). ok
Further comments
Line 162. … was calculated on dry weight basis as …..
Line 214. a concentration
Line 227. Replace “such as such as” with “such as”
Line 234: replace “mineral content” with “toxic elements”.
Line 237: authors should report the legal limit for Pb (referred to meat).
Lines 235-237. Only snail from Biogradska Gora showed a Pb content slightly lower than the legal limit. Authors should underly the difference in Pb content between these snails and the others and should try to explain the reason of this difference. The same for the other toxic elements if there are any differences between sites or species.
Line 240: the Hg concentration of Hg… please correct
Line 242. v.v. ? probably w.w.
Lines 246. To many digits! 4 digits after the comma is too much. As authors can see from their results, the maximum number of significant digits after the comma is 1 or 2. Be consistent also along all the text.
Line 264. Fe plays an important role
Line 268. tissue
3.2. Distributions of metal concentrations in different snail tissues replace metal with element because authors discussed also about Se.
Lines 276-77. Rephrase the sentence. It is grammatically uncorrected and twisted.
The results and discussion section must be rewritten. It is difficult to read, and a bit confusing. for example, authors discuss Se in lines 204-233, then they discuss data about Se on lines 313-315.
In my opinion, first authors should present and compare data obtained in this work between different species and tissues (see section 3.2, that should be integrated with lines 204-…). After that, authors should compare data obtained with data from literature, and, finally, authors should discuss about toxic elements content and legal limit (referring to the new table with toxic elements content in w.w. and legal limit for comparison)
Author Response
"Please see the attachment."

Reviewer 2 Report
This article was revised appropriately.
I recommend accept
Author Response
The authors would like to thank anonymous reviewer for contributing with valuable suggestions and comments.
Reviewer 3 Report
The manuscript can be accepted now in Foods
Author Response

(The authors gave the same response as above.)
